# Effect of the Large and Small T-Antigens of Human Polyomaviruses on Signaling Pathways

**DOI:** 10.3390/ijms20163914

**Published:** 2019-08-12

**Authors:** Ugo Moens, Andrew Macdonald

**Affiliations:** 1Molecular Inflammation Research Group, Department of Medical Biology, Faculty of Health Sciences, UiT The Arctic University of Norway, 9019 Tromsø, Norway; 2School of Molecular and Cellular Biology, Astbury Centre for Structural Molecular Biology, Faculty of Biological Sciences, University of Leeds, Leeds LS2 9JT, UK

**Keywords:** apoptosis, DNA damage response, immune response, interferon, MAP kinase, NFκB, p53, PI3K, protein phosphatases, retinoblastoma

## Abstract

Viruses are intracellular parasites that require a permissive host cell to express the viral genome and to produce new progeny virus particles. However, not all viral infections are productive and some viruses can induce carcinogenesis. Irrespective of the type of infection (productive or neoplastic), viruses hijack the host cell machinery to permit optimal viral replication or to transform the infected cell into a tumor cell. One mechanism viruses employ to reprogram the host cell is through interference with signaling pathways. Polyomaviruses are naked, double-stranded DNA viruses whose genome encodes the regulatory proteins large T-antigen and small t-antigen, and structural proteins that form the capsid. The large T-antigens and small t-antigens can interfere with several host signaling pathways. In this case, we review the interplay between the large T-antigens and small t-antigens with host signaling pathways and the biological consequences of these interactions.

## 1. Introduction

The *Polyomaviridae* family consists of naked viruses with an icosahedral capsid structure. Although originally isolated in mammals, polyomaviruses (PyV) also infect birds and, recently, PyV sequences have also been detected in insects, fish, amphibians, and reptiles. However, it remains to be established whether PyV can actually infect these species [1,2]. The circular double-stranded DNA genome of PyV encodes regulatory and structural proteins, which are expressed in a time-dependent fashion. The regulatory proteins are expressed before the onset of viral DNA replication and are referred to as the early proteins, whereas the structural proteins are synthesized later in the infection cycle and, therefore, are called the late proteins. The early proteins are required for viral DNA replication and transcription, while the late proteins form the capsid [2].

So far, 14 different human polyomaviruses (HPyV) have been described. BKPyV and JCPyV were the first HPyV to be isolated in 1971 and they were named after the initials of the patient [3,4]. In the last decade, 12 novel HPyV have been described: KIPyV [5], WUPyV [6], Merkel cell polyomavirus (MCPyV; [7]), HPyV6 [8], HPyV7 [8], Trichodysplasia spinulosa-associated polyomavirus (TSPyV; [9]), HPyV9 [10], MWPyV [11,12], STLPyV [13], HPyV12 [14], NJPyV [15], and LIPyV [16]. They all encode at least two early proteins: large T-antigen (LT) and small t-antigen (sT), but other early proteins have been detected or may be encoded by the viral genome (Figure 1). Most HPyV produce three late proteins: VP1, VP2, and VP3. BKPyV and JCPyV encode an additional non-structural late protein known as the agnoprotein [17], whereas MCPyV does not seem to express VP3 [18]. HPyV infection is common in the human population. Serological studies have shown a seroprevalence ranging from ~5% for HPyV12, NJPyV, and LIPyV, ~20% for HPyV9 and ≥60% for the other HPyV in the healthy adult population. Moreover, each individual is infected with several HPyVs [19,20]. Primary infection occurs in early childhood, after which the virus establishes a life-long and sub-clinical co-existence with its host [21]. Immunodeficient conditions, immunosuppressive drugs, and pregnancy can lead to reactivation of HPyV and may cause diseases. BKPyV causes polyomavirus-associated nephropathy in renal transplant patients and hemorrhagic cystitis in bone marrow transplants. JCPyV is associated with progressive multifocal leukoencephalopathy and TSPyV is linked to trichodysplasia spinulosa, which is a rare skin disease of severely immunocompromised hosts characterized by follicular distention and keratotic spine formation [22,23]. Despite their name (poly = many and oma = cancers), MCPyV seems to be the only HPyV to induce cancer in its natural host. MCPyV is a major cause in the skin cancer called Merkel cell carcinoma [7,24]. The role of other HPyV, especially BKPyV and JCPyV, in human cancer such as prostate, colorectal, urothelial, and brain cancer is disputed (for recent reviews, see References [25,26,27,28,29]), but some of them can transform cells, including human cells, and the virus or its early proteins LT or/and sT can cause tumors in animal models [27,28,30]. HPyV6 and HPyV7 may be associated with a pruritic rash [31,32], while, so far, no diseases have been associated for the other HPyV.

Viruses, including polyomaviruses, recruit the host cell machinery to favour their replication, and, in the case of oncoviruses, to cause carcinogenesis. One way to take control or perturb cellular processes is by interfering with signaling pathways regulating processes such as DNA replication, the cell cycle, the immune response, transcription, metabolism, DNA repair, cell survival, cell motility, and angiogenesis [33,34,35,36,37]. In the next sections, we review the different pathways that are affected by HPyV and discuss the biological relevance of these interactions.

## 2. Interaction Partners of HPyV LT and sT

One way to explore the impact of HPyV on signaling pathways is to identify which cellular proteins can bind to LT and sT. Several methods such as co-immunoprecipitation, tandem affinity purification coupled to mass spectrometry, GST pull down of in vitro translated proteins, stable isotope labeling by amino acids in cell culture (SILAC)-based pull down, and yeast two-hybrid have been used to identify cellular interaction partners of HPyV LT and sT to understand the function of these proteins [2]. A list of cellular proteins that interact with HPyV LT and sT is given in Table 1. Some of these proteins are part of signaling pathways and will be discussed in Section 3. A special group of proteins that are targeted by HPyV are the protein phosphatases. Because protein phosphatases can interfere with several signaling pathways and targeting them is one of the strategies polyomaviruses use to optimize the host cell for their replication or to transform cells, they will be discussed in more detail in the next paragraphs [38].

The human phosphatome consists of 264 enzymes that, depending on their substrate, include non-protein phosphatases (e.g., phosphoinositide phosphatases), and protein phosphatases. The latter group contains protein tyrosine phosphatases, serine/threonine phosphatases, and dual specificity phosphatases [39]. HPyV viral proteins have been shown to interact with several serine/threonine phosphatases and will be discussed here.

### 2.1. Protein Phosphatase 1 (PP1)

MCPyV sT was found to bind the catalytic subunit of PP1 [40]. The biological relevance has not been examined, but inhibition of PP1 prevents dephosphorylation of retinoblastoma and ensures cell cycle progression [41]. It is known that HPyV drive cells into the S phase in order to facilitate viral genome replication [2]. This may be (partially) achieved by sT-mediated inhibition of PP1, which results in hyperphosphorylation of retinoblastoma.

### 2.2. Protein Phosphatase 2A (PP2A)

PP2A is a phosphoserine/threonine phosphatase that exists as a heterotrimer composed of a structural subunit A, a regulatory subunit B, and a catalytic C subunit [42]. Several isoforms of each of these subunits exist, but it is mainly the B-type subunit that determines substrate specificity, subcellular localization, and catalytic activity of the PP2A holoenzyme [43]. sT of several HPyV have been demonstrated to interact with PP2A and this interaction is mediated by the N-terminal J domain and the C-terminal zinc binding motif of sT [44]. BKPyV sT was originally found to interact with ~56 kD and ~32 kD cellular proteins, which were suggested to represent the scaffolding A subunit and the catalytic C subunit of PP2A, respectively [45]. Later studies showed that BKPyV sT interacts with PP2A Aα [46]. The biological consequences of the BKPyV sT: PP2A interaction have not been studied. JCPyV sT can bind PP2A Aα, the C subunit, and the AC core [46,47,48], and sT inhibits PP2A-mediated dephosphorylation of the agnoprotein, which is a viral protein involved in maturation and release [47]. Different studies showed the interaction between MCPyV sT and PP2A Aβ, and weakly with PP2A Aα, but also with the catalytic subunits PP2Cα and Cβ. The binding of sT to PP2A reduced the catalytic activity of the enzyme [40,46,49,50]. Whereas SV40 sT inhibited binding of B55α, B56α, and B56ε, MCPyV sT excluded only B56α [40]. The biological implications of the MCPyV sT. The PP2A interaction are not known because mutations that abrogate PP2A binding had no effect on sT’s transforming activity [51], nor did it prevent skin hyperplasia in sT transgenic mice [52].

The effect of HPyV sT: PP2A interaction on signaling pathways was investigated for HPyV6 and TSPyV sT. Both sT were shown to bind PP2A-A and PP2A-C subunits when overexpressed in human embryonal kidney HEK293 cells [53,54]. Wild-type, but not non-PP2A binding HPyV6 sT induced phosphorylation and activation of mitogen-activated kinase/ERK kinase 1 and 2 (MEK1/2) - extracellular signal-regulated kinases 1 and 2 (ERK1/2)-c-Jun. The role of sT: PP2A interaction in the life cycle of HPyV6 was not examined, nor was binding of HPyV6 sT to PP2A monitored in HPyV6 infected host cells or tissue. TSPyV sT also activated this mitogen-activated protein kinase (MAPK) pathway, but it was not examined whether this was PP2A-dependent [53,54]. The TSPyV middle T antigen (MT) interacted with PP2A-C and overexpression of wild-type MT, but not a non PP2A-binding MT mutant resulted in enhanced phosphorylation of MEK1/2, ERK1/2, and mitogen-activated protein kinase interacting protein kinase 1 (MNK1). Overexpression of MT had no effect on the phosphorylation of other PP2A substrates such as c-Jun, 4E-BP1, pRb, AKT, SHC, and SRC [55]. Manipulation of the MEK/ERK/MNK1 pathway by TSPyV MT may contribute to the pathogenic properties of this virus because this pathway regulates cell proliferation, and MNK1 plays a role in mRNA translation [56,57].

### 2.3. Protein Phosphatase 4 (PP4)

MCPyV sT was reported to interact with PP4 [40,49,58,59], and this interaction of MCPyV sT and PP4C plays a role in microtubule stabilization [58]. Microtubules are essential components of the cytoskeleton that are important for chromosome segregation, and the control of cell shape and polarized cell motility [60]. Stathmin is a microtubule-associated protein that, in its phosphorylated form, stimulates microtubule assembly, while unphosphorylated stathmin destabilizes microtubules [61]. MCPyV sT was found to increase the total expression levels stathmin, but to decrease its phosphorylation levels, and to promote microtubule destabilization and to stimulate cell motility [58]. The mechanism by which sT increases stathmin levels is not known, but sT-mediated dephosphorylation of stathmin depends on sT binding and interfering with PP4C’s catalytic activity. Another consequence of the interaction between MCPyV sT and PP4 is the induction of cell motility and filopodium formation [59]. Integrin receptors play an important role in the cell motility pathway. They are αβ heterodimers that transmit extracellular signals via mediators, including small GTPases belonging to the Rho family, to proteins that regulate the actin cytoskeleton architecture [62]. Rho family GTPases play also a role in tumor invasion and cancer metastasis as regulators of actin cytoskeletal dynamics [63,64]. Expression of MCPyV sT upregulated the protein levels of stathmin as previously shown [58], but also of cofilin-1, cortactin, and actin-related protein 2/3 complex subunit, which are all involved in actin rearrangements [59]. Ectopic expression of sT expression provoked phenotypic changes in the actin cytoskeleton, which results in the formation of filopodia. Filopodium formation depended on the interaction between MCPyV sT and PP4C, and sT-induced cell motility and filopodium formation required the GTPases RhoA and cell division cycle protein 42 (Cdc42), but not Rac1. sT elicited increased levels of GTP-bound (active) RhoA and Cdc42. sT-triggered activation of Cdc42 and RhoA seems to be mediated by Thr788 and Thr 789 dephosphorylation of β1 integrin by PP4C [59]. In conclusion, the sT: PP4 interaction leads to dephosphorylation of β1 integrin (and maybe other integrins), which then contribute to the cell motility cascade through the small GTPases RhoA and Cdc42. The mechanism by which sT upregulates expression of actin remodeling proteins remains to be determined. MCPyV sT was also shown to stimulate cell motility by upregulating transcriptional levels of A Disintegrin and Metalloproteinase 10 (ADAM 10) [65]. The transcription factors ACAD8, PPARG, and ITGB3BP activate the ADAM10 promoter, and their genes are induced by MCPyV sT [66]. The sT: PP4 association also interferes with the NFκB pathway and will be discussed in Section 3.12.1.

## 3. The Effect of HPyV LT and sT on Signaling Pathways

### 3.1. Phosphatidyl-3-kinase/AKT/Mammalian Target of Rapamycin Pathway

A central signaling cascade that regulates cellular processes such as growth, motility, survival, metabolism, and angiogenesis is the phosphatidyl-3-kinase/AKT/mammalian target of the rapamycin (PI3K/AKT/mTOR) pathway [67]. Perturbed activation of this pathway is observed in various human cancers [68]. Many viruses target this pathway to ensure successful replication, but they may also subvert this pathway to induce cancer [69]. Signaling through the PI3K/AKT/mTOR pathway occurs through ligands that bind to membrane-bound receptor protein tyrosine kinases. This interaction results in autophosphorylation on tyrosine residues. PI3K is then recruited to the membrane by directly binding to these phosphotyrosines. This leads to the activation of PI3K. Activated PI3K converts phosphatidylinositol-4,5-bisphosphate (PIP_2_) into phosphatidylinositol-3,4,5-triphosphate (PIP_3_), which then recruits the protein serine/threonine kinase-3′-phosphoinositide-dependent kinase 1 (PDK1). PDK1 phosphorylates and activates AKT. AKT signaling promotes mTOR activity through inhibitory phosphorylation of tuberous sclerosis complex (TSC) proteins 1/2, which act as mTOR inhibitors. mTOR is a complex that consists of mTORC1 and mTORC2 [70,71].

Earlier studies had shown that SV40 LT, through an interaction with insulin receptor substrate 1 (IRS-1) and sT, and via inhibition of protein phosphatase 2A (PP2A) could activate the AKT pathway [72,73]. It was, therefore, not a surprise that other HPyV could interfere with the PI3K/AKT/mTOR pathway. Following IGF-I stimulation of a JCPyV LT expressing medulloblastoma cell line (BsB8) and a LT-negative medulloblastoma cell line (Bs-1a), a prolonged (>3 h after stimulation) phosphorylation of AKT was observed in the LT-positive cells, whereas a very transient (~30 min) phosphorylation of AKT was observed in the cells lacking LT [74]. Another study showed that stable expression of JCPyV LT in the human colon cancer cell lines HCT116 and SW837 resulted in ~3-fold to ~5-fold increase in invasion and migration compared with empty vector transfected control cells [75]. Microarray analysis identified more than 500 genes involved in cell motility that differentially expressed between the LT expressing HCT116 cells and the control cells. From 43 up-regulated or down-regulated genes involved in migration and invasion, 20 were specifically associated with the AKT pathway. Moreover, phosphorylation levels of AKT were increased in LT expressing cells compared with the control cells and specific AKT inhibitors strongly reduced migration and invasion [75]. Hence, JCPyV LT may promote metastasis by further stimulating the activated AKT pathway. The mechanism by which LT triggered the AKT pathway was not investigated. It should be noted that HCT116 cells contain an activating PIK3CA^H1047R^ mutation [76], which leads to increased AKT activation [77]. However, JCPyV LT further increased the phosphorylation levels, but not the total protein levels of AKT.

The presence of mutations in the PI3K encoding gene (*PIK3CA* gene) and *AKT* gene, as well as the phosphorylation levels of AKT at Thr308 in Merkel cell carcinoma (MCC) tumor samples and cell lines were investigated [78]. Two out of 38 MCC samples had heterozygous mutations in the *PI3KCA* gene, whereas none of the eight cell lines had mutations. The most common mutation that activates AKT, glutamine residue 17 into lysine [79,80], was not detected in any of the MCC tumors or cell lines examined. Immunohistochemistry with phosphoAKT antibodies showed strong or very strong staining for 90% of the samples. However, no significant correlation between phosphoThr308 AKT and MCPyV status in MCC cell lines and MCC tumors was observed [78]. These results suggest that AKT phosphorylation in most MCC is independent of mutations in *PIK3CA* and in *AKT*, and of the presence of MCPyV. The latter indicates that neither LT nor sT are responsible for AKT phosphorylation in MCC. Another study found that the mRNA levels of TSC1, TSC2, and mTOR but not the protein levels, were significantly higher in virus-negative samples compared with virus-positive MCC tumors. AKT phosphorylation at Thr308 was also significantly higher, whereas phosphorylation of Ser473, which is another activation event of AKT, was not statistically significantly different between virus-negative and virus-positive MCCs [81]. Nardi et al. measured AKT phosphorylation at Thr308 and Ser473 in two MCPyV-positive (MKL-1 and MKL-2) and four virus-negative (MCC13, MCC26, UIOS, and MGH-mcc1) MCC cell lines. AKT phosphorylation at both sites was detected in all virus-negative cell lines, but not in the virus-positive cell lines [82]. Taken together, these results indicate that MCPyV does not interfere with the PI3K/AKT/mTOR pathway. This assumption was further confirmed by RNA interference studies. Silencing LT and sT in four MCPyV positive MCC cell lines had no effect on AKT phosphorylation [78]. Despite the high levels of phosphorylated AKT, a role for MCPyV LT and sT in activation of PI3K or AKT in MCC seems unlikely. However, the strong phosphorylation/activation of the AKT cascade in the majority of screened MCC makes this pathway an attractive therapeutic target. The PI3K inhibitor LY294002 abrogated AKT phosphorylation and induced cell cycle arrest and apoptosis in MCC cells [78]. Similarly, rapamycin, which is an mTOR inhibitor, had little effect on MCC cell line survival or proliferation [51]. Whether sT of other HPyV can interfere with the PI3K/AKT/mTOR pathway has not been investigated.

A well–known substrate of the PI3K/AKT/mTORC1 pathway is the eukaryotic initiation factor 4E-binding protein 1 (4E-BP1). 4E-BP1 belongs to a family that contains 4E-BP1, -2, and -3 and exists in one unphosphorylated (4E-BP1α) and three phosphorylated isoforms (β, γ, and δ) with increasing degrees of phosphorylation [83]. In its unphosphorylated or hypo-phosphorylated form, 4E-BP1 binds eukaryotic initiation factor 4E (eIF4E). This interaction prevents assembly of eIF4F onto capped mRNA and, therefore, inhibits cap-dependent translation. In its phosphorylated form, 4E-BP1 dissociates from eIF4E, which, thereby, allows cap-dependent translation. Phosphorylation of human 4E-BP1 can occur at serine residues 65, 83, and 101 and threonine residues 37, 46, and 70, and is mediated by multiple protein kinases [83,84]. 4E-BP1 has been shown to play an important role in cancer [83,84]. Expression of MCPyV LT in HT1080 cells did not promote 4E-BP1 phosphorylation [85], whereas MCPyV and HPyV7 sT enhanced phosphorylation of 4E-BP1δ and, to a lesser extent, 4E-BP1γ at Ser-65. TSPyV and HPyV6 sT had no effect on 4E-BP1 phosphorylation [51,86]. MCPyV sT had no effect on phosphorylation of Thr37, Thr46, and Thr70 [51]. The phosphorylation of these residues by HPyV6, HPyV7, and TSPyV was not investigated [86]. Studies with wild-type and non-PP2A binding MCPyV sT mutants revealed that PP2A was not required for sT-induced 4E-BP1 phosphorylation and was independent of mTOR kinase activity [51]. In a later study by the same group, it was shown that MCPyV sT can provoke phosphorylation of all three threonine residues of 4E-BP1. The authors demonstrated that sT bound to Cdc20 and possibly to the Cdc20 homolog 1 (Cdh1). This activated the CDK1/cyclin B1 complex and CDK1, which resulted in phosphorylation of 4E-BP1 at all sites (Ser83 was not investigated) [87,88]. 4E-BP1 hyperphosphorylation, including at Ser83, was required for MCPyV sT-induced transformation of rodent cells [51,88]. The biological relevance of sT-mediated 4E-BP1 phosphorylation in MCPyV’s lifecyle or MCPyV-induced MCC is not completely understood, but sT-induced hyperphosphorylation of 4E-BP1 can lead to cap-dependent translation, which may contribute to MCPyV’s role in MCC because dysregulated cap-dependent translation promotes tumorigenesis [89]. However, pSer65 4E-BP1 was detected in both virus-negative and virus-positive MCCs. The phosphorylation levels were not quantified, nor was phosphorylation at other phospho-acceptor sites investigated.

### 3.2. Wnt Signalling

The Wnt signaling cascade is a major pathway in cells and aberrant expression is tightly associated with cancer [90]. The Wnt signal transduction pathway can occur in a β-catenin-dependent and β–independent manner. The β-catenin-dependent or canonical pathway depends on the phosphorylation state of β-catenin. β-catenin is part of a multiprotein complex that contains the scaffold protein Axin, the protein kinases glycogen synthase kinase-3 α (GSK3α), and casein kinase 1α (CK1α), as well as the adenomatous polyposis coli (APC) protein. In the absence of a ligand for the Wnt receptor, β-catenin becomes phosphorylated and ubiquitinated and is prone to proteasomal degradation. The Wnt ligand will bind to the Wnt receptor, which is a family known as Frizzled, and its co-receptor low-density lipoprotein receptor-related proteins (LRPs). LRP then becomes phosphorylated by GSK3α and CK1α, which leads to recruiting a disheveled protein (Dvl). Dvl prevents degradation of β-catenin, which leads to accumulation and nuclear translocation. Nuclear β-catenin forms a complex with the transcription activators lymphoid enhancer factor (LEF) and T-cell factor (TCF) and replaces transcription repressors by co-activators such as histone acetyl transferases. This leads to transcription activation of β-catenin target genes [90].

It was demonstrated that JCPyV LT can stimulate transcription of β-catenin target genes and can interact with β-catenin and co-localize in the nuclei [91,92,93]. Transient transfection studies with a luciferase reporter plasmid demonstrated that LT and β-catenin alone increased c-Myc and cyclin D1 promoter activities, while co-expression of LT and β-catenin resulted in a strong synergistic effect [91,93]. In addition, genes encoding cell cycle regulatory proteins and other β-catenin responsive genes (https://web.stanford.edu/group/nusselab/cgi-bin/wnt/target_genes) including genes encoding proteins involved in migration and invasion (e.g., matrix metalloproteinase 7 and Rac1, see below), anti-apoptosis (e.g., survivin), and angiogenesis (e.g., vascular endothelial growth factor) were upregulated by LT. The mechanism by which LT interferes with the β-catenin cascade is not known, but LT may stabilize β-catenin. LT increases β-catenin levels by inhibiting its proteasomal degradation. However, this LT-mediated stabilization of β-catenin is independent of GSK3α [94]. The authors found that, in the presence of LT, β-catenin associates with the cell surface in a Rac1-dependent manner and this resulted in stabilization of β-catenin and activation of Rac1. Alternatively, LT may stimulate nuclear translocation of β-catenin, which is supported by the observation that nuclear localization of β-catenin is more frequent in LT-positive colon cancers compared with LT-negative tumors [93]. Furthermore, LT may retain β-catenin in the nucleus, or a combination of the above-mentioned mechanisms can be imagined. It is clear that JCPyV LT’s ability to interfere with β-catenin contributes to the oncogenic potential of this virus.

### 3.3. Protein Kinase C Pathway (PKC)

Protein kinase C (PKC) is a family of serine/threonine kinases that consists of the members α, βI, βII, γ, δ, ε, η, θ, ζ and ι [95]. Perturbed activity of PKC has been implicated in non-malignant diseases and cancer [95,96,97]. PKCε plays critical roles in cancer development [98], and expression of activated PKCε (serine 729 phosphorylated PKCε) was examined in 11 MCC specimens [99]. Eight of them stained positive with MCPyV LT antibodies and seven of them were also positive for phospho-PKCε. Of the three MCPyV negative MCC samples, only one expressed phospho-PKCε. These results suggest a correlation between PKCε activation and MCPyV positivity in MCC. However, relative few samples were examined and the involvement of MCPyV in PKCε activation remains to be proven. PKCι was upregulated in JCV-infected primary human fetal glial cells [100]. SV40 sT was shown to stimulate PKCλ and PKCζ in a PP2A-dependent manner [101,102]. Whether MCPyV and JCPyV sT operate in a similar manner to activate PKC remains to be established nor is the biological importance in the life cycle of these viruses known.

### 3.4. The Mitogen-Activated Protein Kinase (MAPK) Pathways

The MAPK pathways are involved in processes controlling gene expression, cell division, cell survival, cell death, metabolism, differentiation, and motility. The conventional MAPK pathways consist of a cascade of three consecutive phosphorylation events executed by a MAPK kinase kinase, a MAPK kinase, and a MAPK. There are four different subfamilies of MAPK: extracellular-regulated kinases 1/2 (ERK1/2), c-Jun N-terminal kinases (JNK), p38 MAPK, and big MAPK. The atypical MAPK are not organized in the classical three tier cascade, and include ERK3/4, ERK7/8, and Nemo-like kinase [103].

Infection studies with JCPyV in the human glial cell line SVG-A demonstrated rapid activation of ERK1/2. Increased ERK1/2 phosphorylation could be observed 15 minutes after infection and was sustained for at least 6 hours. At 9 hours post infection, ERK1/2 phosphorylation returned to baseline levels [104,105,106]. Inhibition of ERK1/2 reduced infection of SVG-A cells, but did not affect viral attachment, viral entry, or trafficking, and reduced the early and late promoter activities [105]. The mechanism of JCPyV-induced ERK1/2 activation is not known, but the rapid enhanced phosphorylation of ERK1/2 is consistent with viral attachment and entry events [107] and indicates that ERK1/2 activation occurs prior to expression of LT and sT. Results from another study suggest a negative role of LT on ERK1/2 activation. IGF-I induced ERK1/2 phosphorylation lasted for >1 hour in the LT-negative medullablastoma Bs-1a cell line cells, but ERK1/2 activation subsided after 10 minutes in the LT-positive BsB8 cells [74]. The inhibitory effect of LT may be mediated by LT-induced AKT activation (see 2.1). Activated AKT, in turn, can phosphorylate the MAPK kinase kinase RAF, which leads to the inhibition of the ERK1/2 pathway [108]. Activation of the MAPK ERK1/2 pathway by HPyV may be virus-specific and/or cell-specific because BKPyV did not induce ERK1/2 activation in the human embryonic lung fibroblast cell line HEL-299 and in Vero cells [109].

HPyV may also interfere with other MAPK pathways. One of the substrates of JNK is c-Jun, which is a transcription factor that is part of the dimeric activating protein 1 (AP-1) complex [110]. AP-1 consists of homo-dimeric and heterodimeric complexes of the members c-Jun, JunB, and JunD of the Jun family and c-Fos, FosB, Fra-1, and Fra-2 of the Fos family [111]. JCPyV LT associates with c-Jun and c-Fos and prevents their interaction with the AP-1 DNA binding motif [112]. On the other hand, c-Jun, as well as c-Fos, repressed LT-mediated activation of JCPyV DNA replication and transcription of the late promoter. The functional consequences of the AP-1-LT interaction are not known. For the virus life cycle, it could be a mechanism to favour early viral gene expression because: (i) similar to murine polyomavirus, AP-1 expression was upregulated upon JCPyV infection [113], (ii) AP-1 stimulates transcription from the JCPyV early promoter [114], and (iii) LT usurps AP-1, which prevents viral replication and late gene expression [112]. Moreover, inhibition of AP-1 responsive genes by LT may prevent production of pro-inflammatory cytokines, such as the tumour necrosis factor α (TNF-α), and contribute to immune evasion [115].

The interaction between MCPyV and MAPK pathways has not been intensively studied. Transcript and protein levels of RAF were significantly (*p* = 0.04) higher in MCPyV-positive non-small cell lung cancer samples (*n* = 6) compared with MCPyV-negative non-small cell lung cancer samples (*n* = 10) and an adjacent benign tissue. Moreover, phosphoSer-445 BRAF levels were also significantly higher in the virus-positive specimens than in virus-negative tumours [116]. A larger cohort must be examined and the exact mechanism of MCPyV-induced BRAF expression and phosphorylation must be solved to incontestably establish a role of MCPyV in the activation of the MAPK pathway.

### 3.5. Notch Signaling Pathway

The Notch signaling pathway mediates cell-cell communication because Notch ligands are membrane bound and will interact with the Notch receptors, which are transmembrane proteins. There are four human Notch receptors (Notch1–4), while there are several groups of ligands, including Jagged 1 and 2, and Delta-like proteins. Binding of the ligand to the Notch receptor results in conformational change that exposes Notch to a proteolytic cleavage to release the Notch intracellular domain (NICD). NICD enters the nucleus and will bind to and displace the transcriptional receptor CSL (CBF1/RBPjκ/Su(H)/Lag-1), which results in transcriptional activation of target genes [117,118]. Manipulation of the Notch pathway seems to be a common mechanism used by viruses in carcinogenesis [119]. BKPyV infection of primary human mesothelial cells resulted in increased levels of Notch1, but the biological relevance was not examined [120]. A possible role of the Notch pathway in MCC was investigated by monitoring the levels of Notch1, Notch2, Notch3, and Jagged 1 in MCPyV-negative and positive tumours. Notch3 expression was increased in virus-positive tumours compared to virus-negative ones, while the opposite was found for Jagged 1 [121]. Whether MCPyV proteins are implicated in the upregulation of Notch3 and downregulation of Jagged 1 remains to be investigated. Using tandem affinity purification/mass spectrometry, MCPyV sT, and protein(s) encoded by the early regions of HPyV6 and HPyV7 were shown to interact with Notch2, but the functional consequences of these interactions were not addressed [122].

### 3.6. Hedgehog Signaling

The hedgehog signaling pathway owes its name to the polypeptide ligand, because fruit flies lacking the gene encoding this protein had a phenotype that resembled hedgehogs. In humans, there are three hedgehog ligands: sonic (SHH), Indian (IHH), and desert (DHH) hedgehog. The Patched 1 (PTCH1) receptor forms a heterodimeric complex with the transmembrane protein Smoothened (Smo). In the absence of HH, PTCH1 suppresses Smo by preventing its localization to the cell surface. Binding of HH to the receptor releases this inhibition and allows translocation of Smo to the cell surface. This triggers a signaling cascade resulting in the activation of the DNA binding proteins Gli-1, Gli-2, and Gli-3. In the presence of HH, Gli-1 acts as a transcriptional activator, while Gli-3 is a transcriptional repressor. Depending on the post-translational modifications, Gli-2 can act as a repressor or an activator [123,124]. Expression of SHH and IHH was significantly higher (*p* < 0.001 and *p* = 0.05, respectively) in MCPyV-positive MCCs (*n* = 29) than in MCPyV-negative MCCs (*n* = 21) [125]. Brunner et al. also described higher expression of SHH, IHH, PTCH1, Smo, Gli-1, Gli-2, and Gli-3 in MCC compared to healthy skin and healthy oral mucosa [126]. However, the authors provided no information on the MCPyV status of their MCC samples. The mechanism by which MCPyV may upregulate the HH signaling pathway remains unknown, but virus-induced activating mutations in components of this pathway seem unlikely since mutation analysis showed one silent point mutation in the *SHH-1B* gene and one silent point mutation in exon 5 of the *GLI1* gene in 14 samples [125]. MCPyV-induced activation of the HH pathway may be indirect because of the PI3K/AKT pathway, which can also activate the HH pathway [127].

### 3.7. DNA Damage Response Pathways

DNA damage repair (DDR) pathways will delay or arrest cell cycle progression in cells with damaged DNA. This pathway is controlled by the kinases ataxia telangiectasia mutated (ATM), ATM-related and Rad3-related (ATR), and DNA-dependent protein kinase (DNA-PK) [128,129]. ATM responds primarily to dsDNA breaks, whereas ATR is activated by ssDNA breaks and protects the integrity of replicating chromosomes [130]. Ds breaks trigger autophosphorylation of ATM, which results in the activation of this kinase. Activated ATM phosphorylates Chk2, which subsequently phosphorylates downstream targets, including p53 and H2AX (the phosphorylated form of H2AX is referred to as γ−H2AX). Activation of ATR leads to phosphorylation of Chk1, which, in turn, can phosphorylate several substrates, including p53 and H2AX [128,129]. Phosphorylated p53 triggers expression of p53-responsive genes whose gene products can cause cell cycle arrest, senescence, or apoptosis [131].

Several studies have shown induction of the DDR pathway by JCPyV. JCPyV infection, LT, or sT expression in primary human astrocytes, human neuroblastoma IMR-32, human lung carcinoma H1299 cells, and human glial cell line SVG-A caused genomic instability and DNA damage as shown by aneuploidy/hyperploidy of infected cells, and inhibited nucleotide excision repair (NER) activity. Furthermore, increased phosphorylation of H2AX at Ser-139 (i.e., γ-H2AX) and the ATM and ATR substrates Cdc2, Chk1, Chk2, replication protein A32, and p53 was measured. Elevated levels of DNA repair proteins, including RAD51, p53, Ku70, and phosphorylated Artemis were observed [132,133,134,135]. In addition, BKPyV has been shown to induce the DDR pathway. BKPyV infection of primary human renal proximal tubule epithelial (RPTE) cells resulted in the upregulation of several genes whose products are implicated in the DDR, activation of ATM and ATR, and augmented phosphorylation of their substrates [136,137,138]. Depletion of ATM and/or ATR resulted in decreased viral DNA replication and viral production. In contrast, knockdown of DNA-PK gave ~2-fold increase in viral DNA and viral titer. BKPyV was still able to induce phosphorylation of H2AX in ATM/ATR/DNA-PK triple knockdown cells, which suggests that additional kinases may be involved in the activation of DDR upon BKPyV infection. However, Verhagen and co-workers showed that BKPyV or LT alone were not sufficient to activate ATM and ATR in RPTE cells [139]. The reason for this discrepancy is not known. The ability of JCPyV to induce the DDR pathway may be cell-specific because RAD51 and Ku70 levels were comparable in JCPyV LT negative and LT positive mouse medulloblastoma cell lines [140]. The mechanisms by which JCPyV and BKPyV LT triggers the ATM/ATR DDR pathway and induce chromosome instability are not known, but SV40 LT induced ATM and ATR and this was dependent on the interaction of LT with the mitotic spindle checkpoint kinase Bub1 [141]. The sequence of the LT required for the interaction with Bub1 is conserved in BKPyV and JCPyV LT [141], so that the same mechanism may be operational. Additional mechanisms include LT-mediated upregulation or repression of proteins of the DDR pathway, e.g., JCPyV LT trans-activates the RAD51 promoter [142], whereas sT reduced expression of xeroderma pigmentosum group D protein, which is a protein involved in NER [134]. BKPyV sT interacts with NSe2/Mms21 SUMO ligase, Nse4A, Smc5, and Smc6, a protein complex, which is required for DNA repair [143], but the functional implication remains to be established [122]. JCPyV and BKPyV-induced phosphorylation of DDR proteins may be through sT-mediated inhibition of PP2A. Huang et al. reported that that relatively high concentrations (50–100 µM) of okadaic acid, a PP2A inhibitor, also suppressed NER activity [134]. However, these concentrations of okadaic acid will also inhibit PP1, PP2B, PP4, and PP5 [144], so that the involvement of PP2A in inhibiting NER activity remains to be confirmed. Moreover, the non-PP2A binding sT^P99A^ mutant had no effect on NER activity, which indicates that JCPyV sT can induce chromosome instability in a PP2A-independent manner [134]. Nevertheless, a role for sT-mediated inhibition of PP2A in DNA repair is plausible because it has been demonstrated that dephosphorylation of γ-H2AX by PP2A facilitates DNA repair [145].

Elegant work by the group of Reiss has unveiled the mechanism by which JCPyV LT affects RAD51. They showed that JCPyV LT perturbs homologous DNA repair by interfering with the IGF-1/IGF-IR/insulin receptor substrate 1 (IRS-1) signaling axis and this requires RAD51. The receptor of insulin-like growth factor I (IGF-IR) is a receptor tyrosine kinase that becomes activated by the insulin-like growth factor 1 (IGF-1) and IGF-2. Besides its role in metabolism, this signal transduction pathway is also involved in normal cell growth, DNA repair, regulation of cell-cell adhesion, and cell survival. Compromised activity of the IGF-IR signaling pathway is implicated in cancer [146,147,148]. Normally, hypophosphorylated IRS-1 interacts with Rad51 in the cytoplasm. Upon phosphorylation of IRS-1, Rad51 dissociates, binds BRCA-2, translocates to the nucleus, and gets engaged in DNA repair [149]. JCPyV LT was shown to bind IRS-1 and to induce translocation of cytosolic IRS-1 to the nucleus [150] where it usurps Rad51, which interferes with Rad51′s role in DNA repair [140]. As a result, JCPyV LT reduces the fidelity of DNA repair [140]. IRS-1 contains putative nuclear localization signals [151] so that binding of JCPyV LT does not seem to be required for nuclear import of IRS-1. However, LT may stimulate shuttling of IRS-1 to the nucleus and help to retain it within the nuclear compartment.

The effect of MCPyV on the DDR pathway has also been studied. Infection of osteosarcoma U2OS cells with MCPyV induced the DDR pathway, as shown by enhanced phosphorylation of ATM/Chk1 and ATR/Chk2 [152]. Expression of MCPyV LT in U2OS or cervical carcinoma C33A cells caused DNA damage and elicited phosphorylation of ATM, Chk1 (but not Chk2), H2AX, and p53, and upregulated expression of the p53 target p21^Cip1/Waf1^ [152,153]. The C-terminal region of LT was required for phosphorylation of Chk1 and p53, upregulation of p21^Cip1/Waf1^, and DNA damage. In accordance with elevate p21 levels, cell cycle arrest and inhibition of cell proliferation were observed in LT expressing cells. Full-length LT, but not C-truncated LT, which is typically found in MCPyV positive Merkel cell carcinoma (MCC) tumours, induced the ATR/Chk1/p53 pathway. Full-length had a decreased potential to stimulate cell proliferation and anchorage-independent cell growth compared to truncated LT. These observations may explain why LT is C-terminally truncated in MCC. The authors also showed that MCPyV sT or 57kT failed to increase phosphorylation of Chk1 and Chk2. In another study, the same group showed that MCPyV utilizes host DDR factors for replication of its genome [153]. Additionally, ATM can phosphorylate MCPyV LT at Ser-816. The non-phosphorylatable LT mutant S816A showed reduced growth inhibiting properties and induced less apoptosis compared with wild-type LT [154]. Hence, LT-induced ATM activation may lead to ATM-mediated LT phosphorylation, which affects the functions of LT.

### 3.8. Retinoblastoma-E2F Pathway

The retinoblastoma family or pocket proteins is a family of tumour suppressors that consist of the three members pRb (RB1), p107, and p130 (RB2). They have a pivotal role in controlling cell cycle progression from the G1 to the S phase. During the G1 to the S phase transition, RB1 is converted from its hypophosphorylated form to its hyperphosphorylated form, while p107 and p130 become hyperphosphorylated during the late G1 to S phase [155,156,157]. The retinoblastoma proteins control the S phase cell cycle progression by regulating the transcription of E2F-responsive genes [158]. E2F is a family of transcription factors that consists of eight known members (E2F1–8). RB1 preferentially interacts with the activators E2F-1, E2F-2, and E2F-3a, whereas all RB members can bind the repressors E2F-3b, E2F-4, and E2F-5 [159]. E2F-7a, E2F-7b, and E2F8 also act as repressors but do not seem to bind retinoblastoma proteins [160]. Hypophosphorylated RB1 can repress transcription by direct binding to the activation domain of E2F, which prevents the assembly of the transcriptional pre-initiation complex and by recruiting histone deacetylases [161,162]. Inactivation of retinoblastoma proteins by viral proteins is a common and major mechanism employed by all known human tumour viruses to induce carcinogenesis [163]. It is, therefore, not unexpected that HPyV uses the same strategy. Moreover, inhibition of the pocket proteins will drive the cells into the S phase to facilitate replication of the viral genome.

The LT of BKPyV, JCPyV, WUPyV, MCPyV, and HPyV7 have been shown to interact with the pocket proteins [122,164,165,166,167,168,169,170]. Moreover, BKPyV sT, JCPyV sT, T’165, T’136, and T’135. MCPyV truncated LT and the 57kT variant can also bind retinoblastoma proteins [122,167,168]. However, these HPyV early proteins do not interact with all retinoblastoma family members and they bind with different affinity. BKPyV sT did not bind p107 [122], but was shown to interact with Cdk2 [122], which is a kinase that can phosphorylate pRb [171]. The functional consequences of the BKPyV sT: Cdk2 interaction were not explored. JCPyV LT and the three T’ variants exhibited the highest affinity for p107 and lowest for pRb, whereas sT only binds p107 and p130 [48,170]. The interaction between BKPyV LT and pRb was very weak and estimated to be ~50x weaker than between SV40 LT and pRb [166]. MCPyV truncated LT interacted strongly with RB1, whereas full length LT bound weakly [170]. MCPyV LT did not interact with the p107 and p130 members of the retinoblastoma family, nor did it interfere with p107-induced and p130-induced cell cycle arrest and repression of E2F responsive genes [169,170,172].

A direct interaction between the sT of MCPyV and TSPyV and the pocket proteins has not been demonstrated, but induced expression of TSPyV sT, but not MCPyV sT, which enhanced phosphorylation of pRb in HEK293 cells [173]. Hence, TSPyV sT may play a role in trichodysplasia spinulosa, which is a proliferative cutaneous disease [174]. pRb is a genuine substrate for PP2A [175], so the difference in TSPyV sT and MCPyV sT to induce hyperphosphorylation of pRb may result from their differences in binding and inactivating PP2A isoforms. The PP2A B55α subunit modulates the phosphorylation status of pRb [176] and this subunit was not excluded by MCPyV sT (see Section 2.2). Kazem and co-workers showed that phosphorylated pRb expression in TSPyV LT positive hair follicles was increased when compared to healthy hair follicles from the same patients [177]. Whether LT from this virus interacts with pocket proteins was not investigated.

The biological consequences of the interaction with the pocket proteins was not always investigated. The interaction of JCPyV LT and the three T’ variants inhibited phosphorylation of the pocket proteins and promoted their degradation, whereas association of sT with p107/p130 drove the cells into the S phase and promoted replication of the viral genome [48,178]. BKPyV LT reduced the total levels and the phosphorylation status of all three pocket proteins and increased the amount of free and transcriptionally active e2F detected in kidney fibroblast BSC-1 cells stably expressing LT compared to control cells or cells expressing non-pRb-binding LT mutants [166,179]. McCabe et al. showed that BKPyV LT stimulated the promoter of the *DNA methyltransferease 1* (*DNMT1*) gene and this activation depends on the E2F binding sites in the promoter and on LT’s ability to interact with the pocket proteins [180]. Infection of RPTE cells and primary human prostate epithelial cells with BKPyV resulted in elevated DNMT1 protein levels, which coincided with LT expression. Expression of proliferating cell nuclear antigen (PCNA), cyclin E along with E2F1 and other E2F target genes, was also upregulated and correlated with LT expression [180,181]. Thus, LT may use the pRb: E2F pathway to trigger expression of E2F target genes and this may contribute to virally induce transformation and tumorigenesis. The correlation between DNM1 expression and DNA methylation in BKPyV positive tumours has not been examined, but a significant correlation between JCPyV infection increased DNMT1 expression, and DNA hypermethylation has been found in tumours [182,183,184]. However, not all studies have confirmed a correlation between the presence of JCPyV and methylation (of specific genes) in cancer [185]. MCPyV full-length and truncated LT were able to relieve repression by RB1 of E2F-responsive promoters, but truncated LT was a stronger inducer of E2F-dependent transcription than full-length LT [170]. The biological implications of WUPyV and HPyV LT interaction with the pocket proteins was not investigated. In conclusion, targeting the Rb-E2F pathway may be a common mechanism used by HPyV to stimulate cells entering the S phase, which favours viral replication or may contribute to HPyV-induced tumorigenesis.

### 3.9. p53 Pathway

The tumour suppressor p53 is a transcription factor that binds as a tetramer to DNA in a sequence-specific manner and activates or represses the expression of several hundred target genes [186]. p53 is a key regulator in cell cycle control, DNA repair, cell survival, senescence, autophagy, and angiogenesis [187,188,189]. Mutations in the *TP53* gene are observed in most cancers [190]. Furthermore, inactivation of p53 is a common mechanism in a virus-induced cancer [163]. It is, therefore, no surprise that viral proteins of the HPyV also interact with p53. LT of BKPyV and JCPyV, and the LT’ variants were shown to bind p53 [166,179,191,192,193]. The interaction of BKPyV LT with p53 inhibited p53 and the p53-mediated response to DNA damage. The BKPyV promoter contains p53 binding sites and p53 was shown to repress early promoter activity. Usurping p53 by LT alleviated the inhibitory effect of p53 on the BKPyV promoter [194]. BKPyV-infection of RPTE cells upregulated p53 and downregulated MDM2. An E3 ubiquitin ligase was involved in proteasomal degradation of p53. LT alone was sufficient to reduce MDM2 levels and kept p53 inactive by binding to it [195,196]. p53 repressed JCPyV DNA replication by interacting with LT [192]. Interaction between p53 and the LT of WUPyV was also described. However, the effect of this interaction on viral replication or functions of p53 was not examined [122].

The C-terminal truncated LT that is expressed in MCPyV-positive MCCs lacks the conserved p53-binding domain present in LT of other HPyV and, hence, cannot bind p53 [169]. Surprisingly, full-length MCPyV LT failed to bind p53 as shown by co-immunoprecipitation studies in the human osteosarcoma U2OS cells containing wild-type p53 [197] and transfected with a MCPyV LT expression plasmid [169]. For this study, tagged-LT was precipitated with an antibody against the tag, and then blotted with anti-p53 antibodies for the presence of p53. No reciprocal immunoprecipitation was performed. In another study, Borchert and co-workers ectopically expressed full-length or truncated LT and p53 in wild-type p53 expressing human embryonic kidney cells HEK293 (wild-type p53) and in the p53 null H1299 non-small lung cancer cells [198], and showed that an antibody against p53 could immuno-precipitate full-length, but not truncated LT [170]. However, Forster resonance energy transfer (FRET) studies indicated that LT did not bind p53 directly. LT, but not truncated LT inhibited p53-mediated transcription, indicated that full-length but not truncated LT may interact with a bridging protein that serves as a co-activator in p53-driven transcription. A multimeric complex of MCPyV LT and p53 resembles the situation with the human papillomavirus E6 protein, which forms a complex with E6AP and p53, where neither E6 nor E6AP are separately able to recruit p53, but E6AP renders the conformation of E6 competent for interaction with p53 [199]. A recent study reported that the ectopic expression of C-terminal truncated MCPyV LT in IMR90 lung fibroblasts significantly stimulated transcript levels of p53-responsive genes and increased total protein levels as well as the Ser-15 phosphorylation levels of p53 in a pRb-binding dependent manner, whereas co-expression with MCPyV sT. However, this thwarted the effect of truncated LT [200]. Why there are discrepancies between the findings of Martinez-Zapien et al. and Park et al. is not clear, but the effect of MCPyV LT on p53-mediated transcription may be cell-type specific. The inhibitory effect of MCPyV sT on p53 activity may be explained by sT’s ability to bind and activate the transcription factor MYCL and the histone acetylase complex EP40, and, thus, stimulate MYCL-dependent and EP400-dependent transcription. The sT-MYCL-EP400 complex can transcriptionally regulate the expression of MDM2 and casein kinase 1α (CK1α), which is an activator of MDM4. Both MDM2 and MDM4 induce degradation of p53 [201]. The sT-MYCL/EP400-MDM2/4 connection that represses p53-driven transcription may contribute to the role of sT in tumorigenesis of MCC.

### 3.10. Apoptotic Pathways

Programmed cell death or apoptosis plays an important role in development, aging, tissue homeostasis, and in the defense mechanisms against DNA damage and infections. Failure of apoptosis results in pathological conditions, including developmental defects, autoimmune diseases, neurodegeneration, and cancer [202,203]. Viruses also modulate apoptotic pathways to their advantage to ensure survival of the host cell so that they can complete viral replication, or in the case of tumour viruses to evade apoptosis, which is one of the hallmarks of cancer [204].

HPyV can also interfere with apoptosis. One major mechanism is by neutralizing p53, which is a protein that is central in apoptosis and regulates transcription of numerous genes involved in apoptosis [205]. The interaction between HPyV early proteins and p53 was discussed in Section 3.9. One of the additional targets in the apoptotic pathways targeted by HPyV are the Bcl-2 associated athano-gene proteins (BAG). BAG is a family of co-chaperones that interact with the ATPase domain of heat shock protein 70. Six human BAG proteins (BAG1–6) have been described and they perform a diversity of cellular functions, including apoptosis, differentiation, stress response, proteasomal degradation and autophagy, and cell cycling [206]. Infection of primary human foetal astrocytes with JCPyV resulted in the downregulation of BAG3 expression, whereas ectopic expression of LT in U-87 MG cells reduced BAG3 levels [207]. The authors showed that LT-mediated inhibition of BAG3 requires an AP-2 binding site in the promoter of the *BAG3* gene. JCPyV LT may compete with AP-2 for the same site because the JCPyV LT binding motif 5′-GRGGC-3′ (R = A or G) and the AP-2 consensus site 5′-CCSCRGGC-3′ (S = C or G; [208]) are similar. Alternatively, but not exclusively, JCPyV LT sequesters AP-2 because SV40 LT was shown to interact with AP-2 and to prevent binding to DNA [208]. JCPyV-mediated downregulation of BAG3, which would stimulate cell death. This may seem unfavourable for the successful viral replication. However, another study by the same group showed that the JCPyV LT and BAG3 associate interaction is important for autophagic degradation of LT [209]. Hence, by downregulating BAG3, LT prevents its own degradation. Immunoprecipitation studies showed the interaction between BAG2, BAG3, and BAG5 with MCPyV LT as well as sT, whereas BKPyV sT bound BAG2, and BAG3. However, the biological consequences of these interactions were not tested [122].

Expression of Bcl-2, which is an anti-apoptotic protein that promotes cell survival [210], was significantly (*p* = 0.05) downregulated in MCPyV-positive non-small cell lung cancer samples compared to virus-negative tumours and healthy adjacent tissue, while there were no significant differences in mRNA levels of the pro-apoptotic *Bax* gene between the different specimens [116]. Few samples were compared and the mechanism by which MCPyV may affect transcription of the *Bcl-2* gene was not investigated. However, in another study on a larger cohort of MCC patients (*n* = 116), no statistically significant correlation was found between the presence of the Bcl-2 protein (levels not quantitated) and MCPyV DNA in the tumour samples [211]. Transcriptome analysis of MCPyV-negative and MCPyV–positive MCC tumours could not detect differences in *Bax* and *Bcl-2* levels [212]. The examination of different tumour types may explain the discrepancy between these studies.

Another inhibitor of cell death is survivin, and its expression is upregulated in most cancers [213]. By comparing the transcript level in MCPyV-positive with MCPyV-negative MCC, Arora and co-workers found that ~10% of the 11,500 genes examined were more than three-fold elevated in the virus-positive tumours [212]. A seven-fold upregulation of survivin transcript levels was observed. LT, but not sT, was shown to upregulate expression of survivin [212,214]. Accordingly, RNA interference-mediated depletion of MCPyV early proteins resulted in reduced expression of survivin [172]. Survivin expression is regulated by RB/E2F signaling. Both pRb and p130 can interact with the survivin promoter and repress transcription, whereas E2F members can activate transcription [215]. This suggests that LT induces survivin transcription by usurping pRb/p130, which impairs pRb/p130′s repressing activity on E2F. A LT mutant unable to bind pRb did not stimulate survivin transcription [212]. MCPyV may, thus, through LT-mediated upregulation of survivin, prevent apoptosis. The potent survivin inhibitor YM155 inhibited growth of MCPyV-positive MCC cells in vitro and of xenografts in NOD-SCIDγ mice. Bortezomib, which is another survivin inhibitor, also inhibited MCC cell growth in vitro, but was not active against the MCC xenografts in mice [212]. However, the role of survivin in MCPyV-induced MCC may not be absolutely required because no correlation between survivin expression and MCPyV positivity was found in 64 MCC samples [216]. JCPyV infection of oligodendrocytes and astrocytesic cells also leads to a transiently increased expression of survivin. LT is a likely candidate to upregulate expression of survivin because it was shown to bind and activate the surviving promoter [217].

### 3.11. Ubiquitination-Proteasomal Degradation Pathway

Ubiquitination involves the covalent attachment of a 76 amino acid peptide to target proteins. This labels the proteins for proteolytic degradation by a multiprotein complex known as the 26 S proteasome. The proteasome function is essential for protein homeostasis and influences the regulation of most cellular processes, including cell survival, cell signaling, and cell cycle progression [218,219]. Consequently, dysfunction of the proteasomal system is associated with numerous diseases, including cancer [220,221]. The ubiquitination-proteasomal pathways plays an important role in virus life cycles. It may protect the cell from viral infection by degrading viral proteins, but, on the other hand, viral proteins can hijack this pathway to ensure viral replication and even virus-induced oncogenesis [222].

JCPyV LT binds β-transducin-repeat containing proteins 1 and 2 (βTrCP1 and βTrCP2), which are F-box proteins that are part of the E3 ubiquitin ligase complex SCF^Fbw7^ (Skp1, Cul1, and F box protein; F box and WD repeat domain-containing 7) [223]. This interaction required the DSGHGSS (residues 639–645) motif of LT and phosphorylation of Ser-640 and to a lesser extend Ser-644. The LT: βTrCP interaction did not affect the stability of LT, which suggests that LT is not a substrate, nor were the levels of β-catenin, a βTrCP1 substrate, changed. The effect on other βTrCP1/2 substrates was not investigated. LT of BKPyV (DSGHGSS), MCPyV (DSGTFSQ), and HPyV10 (DSGINSQ) also contain a putative βTrCP1/2 binding motif, whereas TSPyV (DSGFQTQ) and LIPyV (DSGLFTQ) LT have a putative motif, but lack the serine phosphoacceptor site. SV40 LT, which contains the motif DSGHETG, did not interact with βTrCP1 [223]. This suggests that Thr cannot functionally replace Ser and that the LT of BKPyV, MCPyV, and HPyV10, but not of TSPyV and LIPyV may interact with βTrCP. No possible βTrCP binding motifs were detected in the LT of the other HPyV (our unpublished results). The functional implications of the JCPyV LT: βTrCP association remains unexploited, but a JCPyV LT S640A mutant (serine 640 replaced by alanine) impeded viral replication [223]. Moreover, interaction of LT with βTrCP may perturb its involvement in cell cycle regulation and the proteasomal pathway [224], which may lead to a transformation.

MCPyV sT was found to bind and inhibit the E3 ubiquitin ligase complex SCF^Fbw7^. This resulted in stabilization of MCPyV LT, which is a substrate for the E3 ligases SCF^Fbw7^, β-TrCP, and Skp2 [225]. sT-induced stabilization of LT stimulated viral replication and sT also prevented proteasomal degradation of cellular SCF^Fbw7^ targets such as the oncoproteins c-MYC and cyclin E [226]. Furthermore, transformation of rodent fibroblasts in vitro, by sT, was SCF^Fbw7^-dependent [226]. Moreover, MCPyV sT was also shown to interact with two other E3 ubiquitin ligases: Cdc20-anaphase promoting complex [40] and β-TrCP [227], and by targeting E3 ubiquitin ligases, sT stimulated genome instability [225]. All these findings imply that MCPyV sT-mediated inhibition of E3 ubiquitin ligases may be an important contributor in MCPyV-induced tumorigenesis, whereas, during lytic infection, sT may enhance viral replication by stabilizing LT. A recent study, however, failed to detect interaction between sT and SCF^Fbw7^ or β-TrCP, as well as between LT and SCF^Fbw7^. Furthermore, sT was demonstrated to stabilize LT independently of SCF^Fbw7^ [228]. The authors did not observe increased c-Myc levels when sT was expressed. The reason for these discrepancies is presently unknown.

MCPyV and BKPyV sT can form a complex with STIP1 homology and U-box containing protein 1 (STUB1), which is also known as carboxy terminus of Hsc70 interacting protein (CHIP) [122]. This protein has E3 ubiquitin ligase activity and, hence, plays a role in ubiquitin-mediated degradation by the proteasome [229]. This protein has a role in innate and adaptive immunity [230].

### 3.12. Immune Response Pathways

#### 3.12.1. NFκB Signaling Pathway

NFκB is a family of transcription factors that consists of five members that can form homodimers and heterodimers. NFκB is kept in an inactive state in the cytoplasm through interaction with members of the inhibitor of κB (IκB) proteins. Activation of the canonical NFκB pathway is mediated by the IκB kinase (IKK) complex, a heterotrimer of the protein kinases IKKα and IKKβ, and the regulatory subunit IKKγ, which is also referred to as an NFκB essential modulator (NEMO). Activation of IKK depends on phosphorylation of IKKα and IKKβ. The activated IKK complex will then phosphorylate IκB, which results in its degradation. Hence, NFκB is released and translocates to the nucleus, where it affects transcription of NFκB-responsive genes [231]. The transcriptional activity of NFκB is increased by phosphorylation [232]. NFκB target genes encode proteins involved in inflammation, but also the antiviral response [233]. The implication of NFκB in immune responses is well-known [234].

Ectopically expressed MCPyV sT was found to inhibit the NFκB pathway and downregulate expression of NFκB target genes such as *CCL20*, *CXC-9*, *IL-8*, and *TANK* in HEK293 and the virus-negative MCC cell line MCC13 [49]. The authors found that sT inhibited phosphorylation of IKKα and IKKβ, which prevents phosphorylation of IκB and reduces nuclear translocation of NFκB. sT was shown to interact with NEMO and PP4C [49]. Although sT could bind PP4C, interaction with PP4 regulatory subunit 1 (PP4R1) was required for an interaction with NEMO. Therefore, sT associated with a PP4R1-PP4C complex, which mediated binding to NEMO [46]. Taken together, these results suggest that sT stimulates the interaction between NEMO and the protein phosphatase PP4C-PP4R1 complex. Consequently, NEMO-mediated recruitment of PP4C to the IKK complex reduces IKK phosphorylation, with subsequent inhibition of IκB and failure to release, activate (phosphorylate), and translocate NFκB to the nucleus. Studies on MCC biopsies may jeopardize the inhibitory role of sT on NFκB because significantly higher (*p* = 0.034) expression of pSer-536 RelA/p65 subunit of NFκB was observed in MCPyV-positive (*n* = 24) compared to virus-negative (*n* = 17) MCCs. This phosphorylated form resided exclusively in the nucleus [235]. Neither BKPyV sT nor JCPyV sT interacted with PP4R1 [46], which suggested that this unique property of MCPyV sT may contribute to the oncogenic properties of this virus.

The DDR can activate the NFκB pathway also referred to as the “inside-out” or “nuclear to cytoplasm” NFκB signaling [236]. This nuclear initiated NFκB activation occurs via ATM, which phosphorylates NEMO. Upon phosphorylation, NEMO is ubiquitinated and the ATM: NEMO complex is exported to the cytoplasm and NEMO will bind to and activate IKK. IKK, in turn, phosphorylates IκBα, which triggers degradation of IκBα and, as a result, activation of NFκB [236]. White and co-workers showed that JCPyV infection of the human glial cell line SVG-A provoked nuclear transfer of NEMO and that LT caused modification of NEMO [135]. NEMO translocation was most prevalent three days after infection, while, at five days, *p*.i. most NEMO was relocated to the cytoplasm. At three days *p*.i., however, little or no phosphorylated ATM was detected in JCPyV-infected cells. Because inhibition of ATM suppressed viral replication, the authors speculated that stress induced upon JCPyV infection activates ATM, which, in a NEMO-dependent manner, activates NFκB. This transcription factor has previously been shown to stimulate transcription of the early and late viral genes and viral DNA replication [237].

Bromodomain protein 4 (Brd4) is a member of the bromodomain and extra terminal domain family of proteins that recognized acetylated lysine. It acts as a transcriptional and epigenetic regulator by activating transcription factors, transcription elongation factor *p*-TEFb, and chromatin remodeling proteins. Moreover, it can interfere with the NFκB pathway by interacting with IκB [238,239]. Its role in cancer and inflammation is well-established [240,241,242]. Brd4 was found to interact with MCPyV LT, which is bound to the replication origin of the viral DNA. Brd4, in turn, helps recruit replication factor C by direct binding to the largest subunit 1 (RFC1), which facilitates replication of the viral genome [243]. It is not known whether LT of other human polyomaviruses interact with Brd4, but Brd4 stimulates JCPyV early transcription in an NFκB-dependent manner [244].

#### 3.12.2. Innate Immune System

The innate immune system forms an important defense against viruses. Specific pattern recognition receptors (PRRs) serve to identify pathogen-associated molecular patterns (PAMPs) and danger-associated molecular patterns (DAMPs) [245,246]. One of the best understood families of PRRs are the toll-like receptors (TLR), which consists of 10 members recognizing specific PAMPs. TLR2 and TLR4 recognize glycoproteins of the virus particle. TLR3, TLR7, and TLR8 detect viral RNA, and TLR9 is a sensor for hypomethylated dsDNA. Binding of viral dsDNA to TLR9 will activate the NFκB signaling pathway and result in the production of inflammatory mediators [246]. The early regions of BKPyV, JCPyV, KIPyV, WUPyV, and MCPyV downregulated the expression of TRL9 in the B lymphocyte RPMI-8226 cell line [247]. The effect seemed to be cell-specific because the early region of KIPyV and WUPyV had no effect on TLR9 expression in naturally immortalized keratinocytes. For MCPyV, it was shown that LT is responsible for inhibiting TLR9 expression by targeting the transcription factor C/EBPβ. sT could also reduce TLR9 expression to a lesser extent. The C/EBPβ levels did not closely reflect the downregulation of TRL9 by the other HPyV, which suggests that they may use different mechanisms. The mechanism by which sT downregulates TLR9 expression is not known, but it may operate by stabilizing LT [226]. Immunohistochemical staining of MCC biopsies confirmed that decreased expression of TLR9 correlated strongly with MCPyV positivity [248]. The exact role of HPyV-mediated downregulation of TLR9 remains unknown, but it may facilitate establishing a viral infection and/or may help virus-positive tumours to evade the immune system.

#### 3.12.3. Interferon Signaling Pathway

Infection of primary human foetal glial cells and U87MG glioblastoma cells with JCPyV induced the expression of interferon-(IFN) stimulated genes [100]. Another study showed that the LT truncated variant of BKPyV and JCPyV LT, but not their sT, could upregulate the expression of IFN-stimulated genes in mouse embryonic fibroblasts. This induction caused an antiviral state and required signal transducer and activator of transcription 1 (STAT1) activation by LT with an intact RB binding domain [249]. The mechanism by which LT induces phosphorylation/activation of STAT1 via the RB binding motif remains unsolved. BKPyV and JCPyV infection triggers expression of interferon inducible genes in a cell-specific and virus-specific manner. JCPyV infection of primary human astrocytes had no effect on the expression of interferon stimulated genes [250], whereas infection of RPTE cells induced the expression of IFN-stimulated genes [251]. On the other hand, BKPyV infection of RPTE cells failed to induce expression of IFN-stimulated genes [251,252]. Another study found that BKPyV enhanced the expression of the IFN-stimulated genes *ISG15* and *IFIT3* in human endothelial cells [253]. BKPyV infection of microvascular endothelial cells activated the IFN signaling pathway and induced expression of IFN-stimulated genes [252]. The transcription factors interferon regulatory factor 3 (IRF3) and STAT1 were activated upon BKPyV infection of microvascular endothelial cells, but not in RPTE cells [252]. A recent proteomic-based study could not detect changes in protein levels of IFN-stimulated genes in BKPyV-infected RPTE cells after 24, 48, and 72 hours *p*.i. [195]. It is unclear why some cells trigger expression of IFN-stimulating genes in response to BKPyV or JCPyV infection and others do not, but the induced antiviral state may restrict BKPyV or JCPyV replication in the host and enable the establishment of a long-term infection [100]. Alternatively, cells may lack the machinery to detect viral infection or viruses may counteract cellular defense responses through immune-evasion activity. Hence, no expression of IFN-stimulating genes occurs after HPyV infection [195,252].

#### 3.12.4. Cytokines/Chemokines

The primary function of cytokines, including chemokines, is the induction of inflammation and immune responses during viral infection [254,255,256,257]. Infection of cell cultures with JCPyV or BKPyV led to the upregulation of cytokines. For example, infection of human embryonic neural progenitor cells with JCPyV resulted in significant upregulation of the cytokines/chemokines such as CCL5/RANTES, GRO, CXCL1/GROα, CXCL16, CXCL8/IL-8, CXCL5/ENA78, and CXCL10/IP-10 and the chemokine receptor CXCR2. Infection of human cortical collecting duct epithelial cells with BKPyV resulted in downregulation of TNFα expression, but upregulation of the TNF receptors 1 and 2, TLR3, RIG-1, IL-6, IL-8/CXC8, CCL5/RANTES, CCL2/MCP-1, and CXCL10/IP-10 [258,259,260]. The mechanism(s) by which BKPyV and JCPyV alter the expression of these proteins has not been investigated, but BKPyV LT expression coincided with upregulation of these proteins so that a role for LT cannot be excluded [258,259]. Comparing the expression levels of 85 cytokines in non-infected and in BKPyV infected human kidney epithelial cells did not reveal any significant changes, even over time [136]. MCPyV can also modulate the expression of cytokines. Transcriptomic analysis of BJ human foreskin fibroblasts stably expresses a C-terminally truncated variant of MCPyV LT (residues 1–339) or this truncated LT plus sT resulted in increased expression of several cytokines and chemokines, including IL-1β, IL-6, IL-8, and CXCL1 [261]. Another chemokine whose expression is upregulated by MCPyV is chemokine (C-C motif) ligand 17/thymus and activation-regulated (CCL17/TARC). Full-length and truncated MCPyV LT, but not sT, enhanced the CCL17/TARC promoter activity and increased protein levels [262]. The presence of a putative E2F binding site in the CCL17/TARC promoter suggests that LT (partially) triggers CCL17/TARC expression through the activation of E2F by relieving repression by RB. The chemokine-like proteins prokineticins possess angiogenic and immunoregulatory activities and may, therefore, be implicated in cancer [263]. MCPyV-positive MCCs had higher prokineticin-2 transcript levels than the virus-negative tumours [264]. The mechanism by which MCPyV may affect prokineticin-2 expression has not been scrutinized, nor has the role in the life cycle or involvement in MCC been explored.

### 3.13. Nuclear Receptor Signaling Pathway

Nuclear receptors are a diverse family of intracellular receptors that also act as transcription factors. Their ligands can pass the plasma membrane and include steroid hormones, thyroid hormone, vitamin D3, retinoic acid, and fatty acid metabolites. Ligand binding, posttranslational modifications, and recruitment of co-activators result in activation of nuclear receptors. They bind as monomers, homodimers, or heterodimers to specific DNA sequences, which directs transcription of their target genes. Nuclear receptors regulate cellular processes such as cell proliferation, development, metabolism, inflammation, tissue homeostasis, apoptosis, and reproduction [265,266]. The implication of nuclear receptors in cancer is well-known as perturbed nuclear receptor signaling, which leads to aberrant gene expression. Nuclear receptors also play a role in regulating angiogenesis and inflammation [267,268,269].

Co-expression of BKPyV LT and sT stimulated oestrogen receptor-mediated transcription. However, when expressed separately, only LT induced oestrogen receptor-mediated transcription [270]. The mechanism by which LT/sT regulated the transcriptional activity of the oestrogen receptor has not be determined, but did not require interaction between LT and the receptor. Glucocorticoids, progesterone, and oestrogen stimulated the BKPyV promoter, enhanced viral early and late gene expression, and increased the virus yield [271]. Whether LT stimulates transcription directed by the glucocorticoid and progesterone receptors has not been investigated, nor has the effect of other HPyV on nuclear receptor signaling been examined.

### 3.14. Phospholipid Signaling Pathways

Sphingolipids are bioactive (phospho)lipids that act as important regulators of cellular processes such as proliferation, cell survival, differentiation, migration, autophagy, and immune responses [272]. These molecules can affect signaling pathways, including PKCζ, NFκB, PI3K/AKT, and JNK and dysregulation of sphingolipid metabolism contributes to tumorigenesis and metastasis [272,273,274]. Sphingosine kinases 1 and 2 (SK1 and SK2) are important metabolic enzymes in the formation of sphingosine-1-phosphate [272], and expression levels of these protein kinases are frequently elevated in many cancers [274]. Bhat and co-workers reported that the transcript levels of SK1 and SK2 were significantly higher in MCPyV-positive MCC cell lines compared to MCPyV-negative cells and that ectopic expression of truncated variants of LT or of sT in human lung fibroblasts MRC-5 cells resulted in increased SK1 and SK2 mRNA concentrations [275]. The activation levels of SK1/2 were not investigated, nor is the mechanism by which LT and sT upregulate SK1/2 transcript levels known. Because truncated LT variants lacking their DNA binding domain still increased SK1/2 mRNA levels, direct binding of LT to the promoters of the *SPHK1* and *SPHK2* genes can be excluded. SK1 becomes activated by Ser-225 phosphorylation and this site is dephosphorylated by PP2A [272]. Therefore, MCPyV sT may prolong SK1 activation by inhibiting PP2A-mediated dephosphorylation. Inhibition of SK1 and SK2 attenuated MCC tumour growth [275], so it may seem that MCPyV LT and sT upregulate expression of SK1 and SK2 to promote proliferation of the tumour cells.

### 3.15. Metabolic Pathways

Viruses can alter the host metabolism for their own benefit to promote optimal viral replication conditions [276]. Human tumour viruses have also been shown to alter glucose metabolism in the tumour cell, which is one of the hallmarks of cancer [163,277]. Studies with JCPyV have shown the importance of normal glucose metabolism for efficient replication. Glucose starvation of human glioblastoma U-87MG cells transiently expressing JCPyV LT and mouse medulloblastoma BsB8 cells stably expressing JCPyV LT and sT, reduced LT protein levels in the glioblastoma cells and both LT and sT levels in the medulloblastoma cells. Activation of 5′-adenosine monophosphate activated protein kinase (AMPK) caused repression of LT expression, whereas inhibition of AMPK restored LT protein levels under glucose deprivation in both cell types. On the other hand, JCPyV LT was shown to suppress 5′-adenosine monophosphate activated protein kinase (AMPK) and to upregulate expression of the glycolytic enzyme transaldolase-1 during glucose deprivation [278]. The interplay between AMPK and LT enabled JCPyV to maintain the cells in the G2 phase during glucose deprivation, which, thereby, prevents the cells from dying. A similar function was found for SV40 sT, which activated AMPK in a PP2A-dependent manner under conditions of glucose starvation and this reduced the rate of cell death [279]. The mechanism by which JCPyV LT regulates AMPK and whether JCPyV sT can activate AMPK remains to be resolved. Doxocyclin-induced expression of MCPyV sT in human lung fibroblast IMR90 cells stimulated transcript levels of glycolytic genes, including hexokinase 2, glucose transporters GLUT1 and GLUT3, transporter for lactate and pyruvate SLC16A1 (MCT1), and the transcription factors MLX and MLXIP, which regulate the transcription of genes encoding glycolytic enzymes [66]. sT-induced expression of SLC16A1 (MCT1) is partially mediated by NFκB [66], but other pathways are likely involved in sT-induced upregulation of metabolic genes. MCPyV-mediated upregulation of glycolysis may contribute to the oncogenic potential of this virus.

## 4. Conclusions and Future Research Directions

Most human polyomaviruses establish a life-long persistent, but harmless infection in healthy people. However, they may cause diseases in immunocompromised individuals. Only one HPyV and MCPyV is recognized as a human oncovirus, whereas BKPyV and JCPyV are possibly oncogenic viruses [280]. Currently, no specific vaccines or efficient drugs against these viruses exist [2]. Human polyomaviruses, like other viruses, reprogram the intracellular environment of their host cell to create an optimal environment for their replication or, in some cases, to transform the host cell into a tumour cell. One mechanism by which viruses interfere with cellular processes is by targeting signaling pathways. The LT and sT of HPyV are multifunctional proteins that are not only required for virus replication, but they are also pivotal in modulating multiple cellular signaling pathways, which we described in this review and summarized in Figure 2A,B. Through their association with components of signaling pathways, HPyV can affect cellular processes such as cell cycle, cell survival, DNA damage repair, transcription, and translation and evade immune surveillance. All these processes will support viral replication. However, these functions of LT and sT may also contribute to oncogenesis, as outlined in this review. The mechanisms by which these viral proteins interfere with different signal transduction pathways and the biological implications are not fully understood. As outlined above, HPyV encode LT and sT with the potential to modulate different signaling pathways that are known to be involved in cancer when they are perturbed. However, only MCPyV is firmly associated with cancer, while a probable implication of BKPyV and JCPyV in human tumours is suggested [27,28,281]. The reason for the differential oncogenic potential of HPyV is not fully understood. In vitro studies and transgenic animal models have shown that MCPyV sT is more oncogenic than LT, while the opposite is true for BKPyV and JCPyV, whose LT seems to be more oncogenic than sT [282]. BKPyV and JCPyV LT share 83% amino acid identity, whereas BKPyV LT (resp. JCPyV LT) and MCPyV LT are only 49% (resp. 48%) identical. Similarly, the sT of BKPyV and JCPyV are 78% identical, whereas BKPyV sT (resp. JCPyV sT) and MCPyV sT share 35% (resp. 33%) identity. These differences in LT and sT sequence may explain why these proteins bind different cellular targets (Table 1) and, hence, have different oncogenic properties. Moreover, the affinity for the cellular proteins can be different, as was illustrated for, e.g., PP2A (Section 2.2) and the retinoblastoma family members (Section 3.8). Other factors that may explain the distinct oncogenic properties of HPyV is the cell tropism, state of the viral genome (integrated or episomal), expression levels of LT and sT, and variants of LT (e.g., truncated MCPyV LT in MCC). This is an important area that needs further investigation. More research is also required to understand the exact molecular basis for the effect of LT and sT on signaling pathways. Comparative proteomic studies between control cells and LT or/and sT expressing cells can identify signaling proteins whose expression is affected by these viral proteins. Similarly, phospho-proteomics of HPyV sT expressing and control cells can be performed to identify putative targets of protein phosphatases that are modulated by HPyV sT and proteomic studies. The biological consequence of LT/sT interaction partners needs to be further explored and phospho-specific antibodies can be used to identify protein kinases and their substrates that become activated upon expression of LT and/or sT. This additional knowledge can then be used to develop therapeutic drugs that prevent LT and sT to perturb signaling pathways. The intimate co-existence between the virus and its host and the necessity of these signaling pathways for normal cellular function forms a big challenge in designing drugs that affect the virus but not the host.

## Figures and Tables

**Figure 1 ijms-20-03914-f001:**
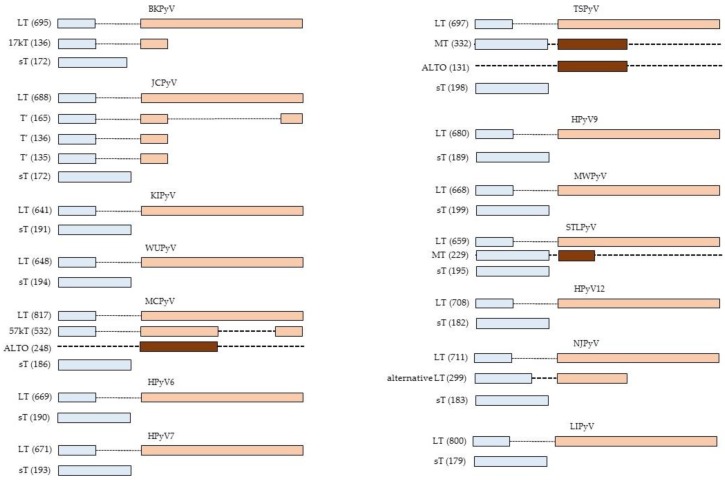
Proven and putative early proteins are encoded by the different HPyV. The number in parenthesis is the number of amino acid residues in the protein. The dashed lines represent non-coding regions, while the colored boxes depict the distinct areas that compose the protein. Part of the N-terminal region of LT and sT has the same amino acid sequence and, therefore, the same color was used. The proteins are not drawn to exact scale. LT = large T-antigen. sT = small T-antigen. MT = middle T-antigen. ALTO = alternative LT open reading frame.

**Figure 2 ijms-20-03914-f002:**
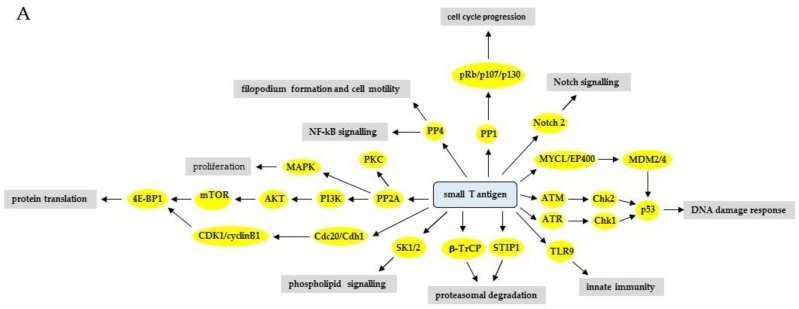
Interaction between HPyV small T antigen (**A**) and large T antigen (**B**) and different signaling pathways. See text for details.

**Table 1 ijms-20-03914-t001:** Cellular proteins identified to interact with the early proteins of HPyV [2].

Virus	LT	sT	T’135	T’136	T’165
BKPyV	p53, pRb	ABCA13, ANKRD30B, ATP2A2, BAG2, BAG3, BAG5, cathepsin BCCND3, CD44, CDK2, CDKN1A, CNP, CSRP1, DnaJC7, DP1, E2F3, E2F4, E2F5, GLIPR2, HSDL2, Hsp70, HSPA4L, HSPBP1, NAGK, Nse2/Mms21, PCNA, PP2CA, PP2R1α, PPM1B, RB1, RBL1, RBL2, SCCPHD, SEC61B, SMC5, SQRDL, SRP9, SRRM2, STUB1, TGFBI			
JCPyV	AP1, BAG3, BRN1, β-catenin, CEBP, Hsp70, IRS-1, LEF1, NF2, Oct6, pRb, p53, Purα, SKP1, YB-1	PP2Cα	Hsp70, pRb	pRb	pRb
WUPyV	p53, RB1				
MCPyV	ABCA13, ABCD3, AP2A1, ATM, BAG2, BAG3, BAG5, Brd4, CREBBP, CK2β, DDX24, DnaJC7, DP1, E2F3, E2F4, EMD, FAM71E2, GTF3C1, HDLBP, Hsp70, IκBIP, KPNA2, KPNA3, KPNA4, MAP4, MED14, P4HA3, PGAM5, PIP4K2 β, PP2AR1α, PTRF, RB1, RTN4, SALL2, SDPR, SGPL1, SRP14, SRPRB, STUB1, TCEB1, TRIM38, TSPYL1, Vam6p, VAPA, VAPB, VPS11, USMG5	ABHD12, ACBD3, ADAM9, AIP, ANKRD13A, ATP2A2, BAG2, BAG3, BAG5, cadherin 1, CCHC, CD44, CDC20, CDH, CNP, COPG2, DnaJA1, DnaJB4, DnaJC7, EFEMP2, eIF4EBP1, emerin, Fbxw7, Hsp70, IGF2R, IκBIP, LOX, MBOAT7, MMP14, MPZL1, MTCH2, myoferlin, NEMO, Notch2, NSD1, P4HB, PDGFRβ, PGRMC2, PRAF2, PPP2CA, PPP2CB, PPP2R1A, PP2R1B, PP4R1, PPM1A, PPM1B, PPM1G, PSMC2, PSMC3, PSMC4, PTTPG1IP, Rab18, RNH1, RPL21, RPs27L, SPARC, SQRDL, SRPRB, STUB1, SURF4, TIMM8A, TMEM165, TMX3, TOLLIP, USMG5, VKORC1, YAP1			
HPyV6	p53, RB1	PP2Cα, PP2R1α			
HPyV7	p53, RB1				
TSPyV	p53, RB1	PP2Cα			
MWPyV	pRb	PP2R1α

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
