# Peer review of "Effect of the Large and Small T-Antigens of Human Polyomaviruses on Signaling Pathways"

_ijms, 2019, doi:10.3390/ijms20163914_

Round 1

Reviewer 1 Report

This review presents information obtained so far on the large and small T-antigens of the so far known human polyomaviruses.

It is ambitiously and cautiously written and gives a broad overview of the field. It describes interaction partners of HPyV small and large T and how they may differ between the different viruses. It covers different signalling pathways, from the intracellular ones important for gene regulation, transformation, cell cycle control, DNA repair, apoptosis etc to those of the immune response and those of metabolic pathways and more. 

Much of the information is focused on small and large T-antigens of BKPyV, JCPyV and MCPyV, where most knowledge is acquired at this time point. When information is available it is also given  for as many of the other HPyVs as possible. Comparisons are done accurately, between the viruses and these are very informative. Many relevant and important references are included.

The table and the two figures are very informative.

I noticed two typographical errors one of page 9, line 334, should read ... studied. The other on page 10, line 376 should read ...seem.  There could be more, but in general the text is clear and easy to follow.

The text could after text checking be published in its present form.

Author Response

We thank the reviewer for the very positive evaluation of our manuscript. We have corrected the typos and double-checked the text once more for spelling mistakes but could not find any additional ones.

Once again, our sincere thanks for the encouraging evaluation.

Reviewer 2 Report

The review is very well organized, complete, helpful for scientists working in Polyomaviruses field. Figures and table are impressive.

I believe the manuscript might be accepted as it is.

Author Response

We thank the reviewer for the very positive evaluation of our manuscript.